# Structure of human lysosomal acid α-glucosidase–a guide for the treatment of Pompe disease

Véronique Roig-Zamboni[1], Beatrice Cobucci-Ponzano [2], Roberta Iacono [2], Maria Carmina Ferrara [2], Stanley Germany[1], Yves Bourne[1], Giancarlo Parenti[3,4], Marco Moracci [2,5] & Gerlind Sulzenbacher [1]

Pompe disease, a rare lysosomal storage disease caused by deficiency of the lysosomal acid α-glucosidase (GAA), is characterized by glycogen accumulation, triggering severe secondary cellular damage and resulting in progressive motor handicap and premature death. Numerous disease-causing mutations in the *gaa* gene have been reported, but the structural effects of the pathological variants were unknown. Here we present the high-resolution crystal structures of recombinant human GAA (rhGAA), the standard care of Pompe disease. These structures portray the unbound form of rhGAA and complexes thereof with active site-directed inhibitors, providing insight into substrate recognition and the molecular framework for the rationalization of the deleterious effects of disease-causing mutations. Furthermore, we report the structure of rhGAA in complex with the allosteric pharmacological chaperone N-acetylcysteine, which reveals the stabilizing function of this chaperone at the structural level.

[1] Centre National de la Recherche Scientifique (CNRS), Aix-Marseille Univ, AFMB, 163 Avenue de Luminy, 13288 Marseille, France. [2] Institute of Biosciences and Bioresources, National Research Council of Italy, Via P. Castellino 111, 80131 Naples, Italy. [3] Telethon Institute of Genetics and Medicine (TIGEM), Via Campi Flegrei 34, Pozzuoli 80078, Naples, Italy. [4] Department of Translational Medical Sciences, Federico II University, Via Pansini 5, 80131 Naples, Italy. [5] Department of Biology, Federico II University, Complesso Universitario di Monte S. Angelo, Via Cintia 21, 80126 Naples, Italy. Correspondence and requests for materials should be addressed to G.S. (email: Gerlind.Sulzenbacher@afmb.univ-mrs.fr)

Pompe disease (also known as glycogen storage disease type 2 or acid maltase deficiency) is an autosomal recessive disorder caused by mutations in the gene that encodes the hydrolase acid α-glucosidase (GAA), member of glycoside hydrolase family GH31[1], and involved in the lysosomal breakdown of glycogen. Functional deficiency of GAA results in lysosomal accumulation of glycogen and cellular damage in all tissues, particularly cardiac and skeletal muscle[2,3]. Pompe disease is characterized by a broad phenotypic spectrum that ranges from a slowly progressing late-onset phenotype to a devastating classical infantile-onset, but in all cases, progressive muscle hypotonia and loss of motor, respiratory and cardiac functions leads to respiratory failure[2,4]. Recombinant human GAA (rhGAA) produced in CHO cells has been approved in 2006 for enzyme replacement therapy (ERT) to treat Pompe disease and has proven to be beneficial for patient's survival and to stabilize the disease course[5–8]. Since then, ERT is the only approved treatment for Pompe disease, but despite the clinical benefits, patient's response is very variable and the efficiency of the treatment is limited by insufficient targeting and uptake in muscle tissues, immunogenic reactions and build-up of autophagic compartments in myocytes[9–12]. Hence, the quest for alternative therapeutic strategies based on different approaches and rationale has become compulsory and pharmacological chaperone therapy (PCT) has been proposed as an alternative or complementary approach to ERT[4,13,14]. PCT is based on the concept that small-molecule ligands may act by blocking conformational fluctuations of a partially misfolded protein, rescuing its functional state and allowing escape from the endoplasmic reticulum-associated protein degradation (ERAD) machinery (Fig. 1)[15]. Hundreds of disease-causing mutations in the *gaa* gene have been identified, including insertions, deletions, splice site, nonsense, and missense mutations (Supplementary Table 1). Missense mutations result in production of full-length GAA, likely to not fold as efficiently as the stable wild-type enzyme, and patients affected by these mutations are thus potential candidates for PCT[16]. While not more than ~10–15% patients are estimated to be amenable to PCT[16], it has been shown that pharmacological chaperones act as enzyme enhancers when co-administered with rhGAA, by favoring enzyme delivery, stability and maturation[17], making this PCT independent from the type of mutation carried by patients.

Here we report the high-resolution structures of mature rhGAA and its complexes with the GAA inhibitors acarbose, 1-deoxynojirimycin (DNJ), N-hydroxyethyl-DNJ (NHE-DNJ) and with the allosteric pharmacological chaperone N-acetylcysteine[18]. These structures give insight into substrate recognition, support at the molecular level the action of currently known pharmacological chaperones, and provide a molecular framework for the rationalization of mutations in clinical isolates of individuals affected by Pompe disease.

## Results

**Structural overview.** GAA is synthesized as a 110 kDa glycoprotein, which is targeted to the lysosome via the mannose-6-phosphate receptor and undergoes in the late endosomal/lysosomal compartment a series of proteolytic and N-glycan processing events to yield a mature active form composed of four tightly associated peptides[19,20]. Because crystallization of the commercial Myozyme® precursor form of rhGAA (Q57-C952) did not afford crystals diffracting beyond ~7 Å and protein disorder predictors[21] indicated the presence of disordered peptide regions, we performed in situ proteolysis with α-chymotrypsin to remove putative flexible surface loops hampering formation of productive crystal contacts. The proteolytic treatment yielded a polypeptide of ~5 kDa lower mass than the rhGAA precursor (Fig. 2a), which crystallized readily. This allowed us to collect the diffraction data extending to 1.9 Å and solve the structure of rhGAA by molecular replacement (Table 1). The proteolytically digested form had activity comparable to that of the precursor ($2.34 \pm 0.06$ and $2.26 \pm 0.16$ U mg$^{-1}$ for the precursor and mature

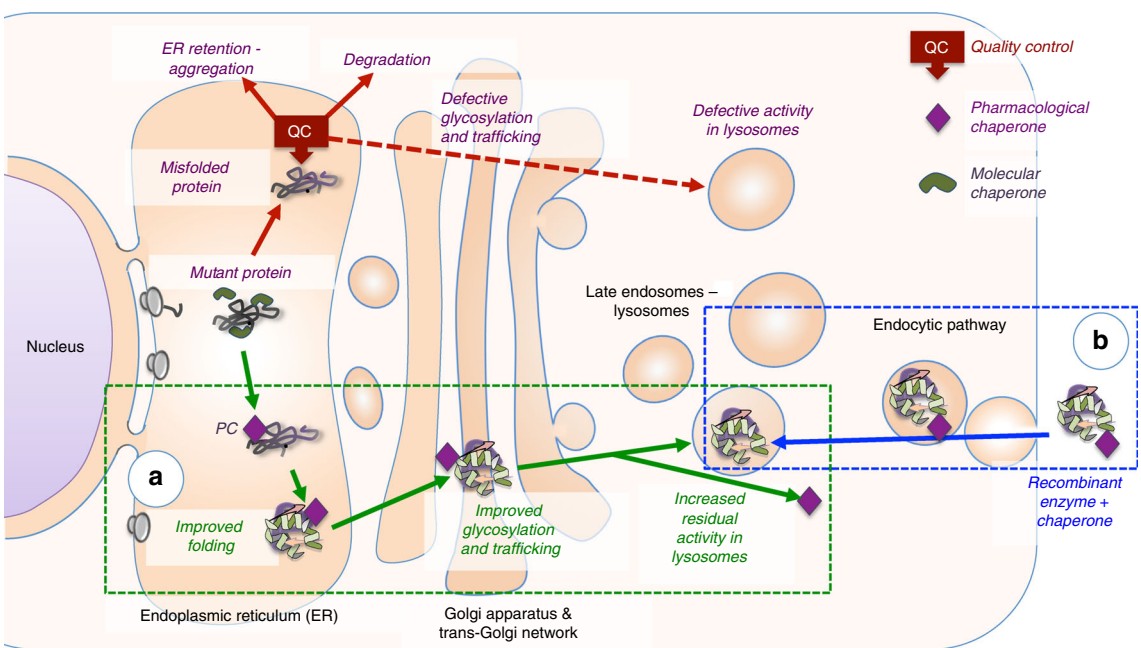

**Fig. 1** The effect of pharmacological chaperones on misfolded lysosomal enzymes and on recombinant enzymes used in ERT. Lysosomal enzymes are assisted by molecular chaperones during synthesis. Mutated enzymes fail to fold correctly and are intercepted by the quality control (QC) system of the endoplasmic reticulum (ER). **a** Pharmacological chaperones favor proper folding of mutated enzymes, prevent their recognition by the quality control system and stabilize the enzyme during transport to their destination. **b** Pharmacological chaperones can enhance the effect of recombinant enzymes administered in ERT by favoring trafficking to lysosomes and increasing enzyme stability

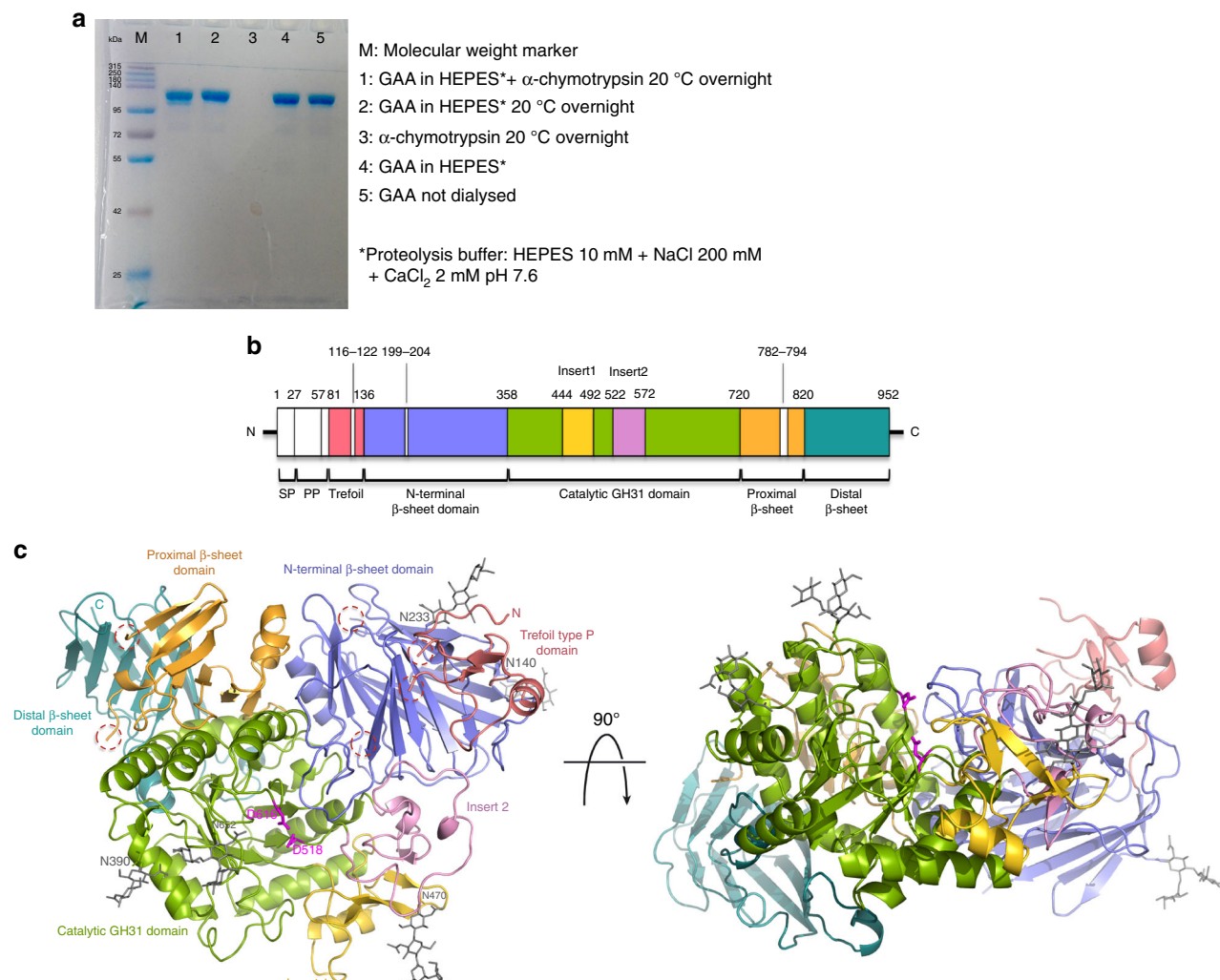

**Fig. 2** Structure of mature rhGAA. **a** Proteolytic treatment of rhGAA. **b** Schematic representation of the sequence of GAA. Myozyme® rhGAA used in ERT starts at residue Q57. Domains corresponding to the rhGAA structure are colored as in **c**, with the regions removed by treatment with α-chymotrypsin colored in white. **c** Cartoon representation of the structure of rhGAA consisting of the trefoil type-P domain (salmon), the N-terminal β-sheet domain (slate), the catalytic GH31 (β/α)$_8$ barrel domain (green) with insert I (gold) and insert II (pink), and the proximal (orange) and distal (teal) β-sheet domains. Catalytic residues (magenta) and glycan chains (grey) are depicted as sticks. Dashed lined circles highlight peptide portions removed by proteolysis with α-chymotrypsin before crystallization. (SP, signal peptide; PP, propeptide)

forms, respectively), indicating that the α-chymotrypsin treatment did not alter the functionality of rhGAA.

The crystal structure of rhGAA corresponds to the mature form of human placental GAA and rhGAA[20] (Fig. 2b), suggesting that treatment with α-chymotrypsin afforded removal of protease-labile regions (Q57-T80, G116-M122, R199-A204 and A782-R794) in a manner comparable to the proteolytic maturation occurring in vivo. The overall architecture of rhGAA is similar to that of the human glycoside hydrolase family GH31[1] homologues, the intestinal brush-border enzymes maltase-glucoamylase (MGAM)[22,23] and sucrase-isomaltase (SI)[24] (Fig. 2c and Supplementary Fig. 1). A N-terminal trefoil Type-P domain is followed by a β-sheet domain, the catalytic (β/α)$_8$ barrel bearing two inserts after β-strands β3 (insert I) and β4 (insert II), and proximal and distal β-sheet domains at the C-terminus. The precursor form of rhGAA contains ~14 kDa of carbohydrate and seven glycosylation sites have been predicted and characterized[20]. We observed glycan structures of various lengths for five of them in the crystal structure, notably at N140, N233, N390, N470 and N652 (Fig. 2c and Supplementary Fig. 2). We detected residual difference electron density at N882, insufficient to model a glycan

structure, and did not observe difference electron density near N925.

**Active site and inhibitor binding.** The narrow substrate-binding pocket of rhGAA is located near the C-terminal ends of β-strands of the catalytic (β/α)$_8$ domain and shaped by a loop from the N-terminal β-sheet domain and both inserts I and II (Fig. 2c). Family GH31 enzymes perform catalysis via a classical Koshland double displacement reaction mechanism with retention of the anomeric carbon configuration in the product. From early characterization of the catalytic site of GAA[25] and by homology with mechanistically dissected members of family GH31[26,27], the catalytic nucleophile and acid/base can be assigned to D518 and D616, respectively. To get insight into the interaction of rhGAA with the documented pharmacological chaperone 1-deoxynojirimycin[16,28] and its derivative N-hydroxyethyl-deoxynojirimycin[29] (DNJ and NHE-DNJ, respectively), we soaked rhGAA crystals with these active site directed iminosugar inhibitors and obtained for both complexes diffraction data extending to 2.0 Å resolution (Table 1). DNJ binds in a $^4C_1$

**Table 1 Data collection and refinement statistics (molecular replacement)[a, b]**

|  | rhGAA | rhGAA-NAC | rhGAA-DNJ | rhGAA-NHE-DNJ | rhGAA-acarbose |
|---|---|---|---|---|---|
| *Data collection* | | | | | |
| Space group | $P2_12_12_1$ | $P2_12_12_1$ | $P2_12_12_1$ | $P2_12_12_1$ | $P2_12_12_1$ |
| Cell dimensions | | | | | |
| *a, b, c* (Å) | 98,102,129 | 97,102,129 | 97,103,129 | 97,102,128 | 97,103,128 |
| Resolution (Å) | 48.83-1.90 | 47.62-1.83 | 53.80-2.00 | 47.47-2.00 | 51.49-2.45 |
|  | (1.93-1.90) | (1.86-1.83) | (2.04-2.00) | (2.04-2.00) | (2.54-2.45) |
| $R_{merge}$ | 0.103 (1.711) | 0.072 (0.569) | 0.159 (0.549) | 0.134 (0.602) | 0.188 (0.885) |
| $R_{pim}$ | 0.050 (0.823) | 0.040 (0.311) | 0.125 (0.426) | 0.091 (0.421) | 0.116 (0.554) |
| CC (1/2) | 0.992 (0.661) | 0.999 (0.859) | 0.965 (0.675) | 0.992 (0.722) | 0.986 (0.644) |
| $I/\sigma I$ | 13.7 (1.2) | 17.8 (3.3) | 6.2 (2.0) | 8.4 (2.3) | 8.0 (1.9) |
| Completeness (%) | 100 (100) | 99.6 (92.0) | 95.3 (95.5) | 99.5 (100) | 94.7 (97.0) |
| Redundancy | 9.0 (9.2) | 7.5 (7.1) | 3.1 (3.0) | 4.5 (4.2) | 5.3 (5.2) |
| Wilson B (Å$^2$) | 28.51 | 21.25 | 14.22 | 16.73 | 25.97 |
|  | | | | | |
| *Refinement* | | | | | |
| Resolution (Å) | 48.83-1.90 | 47.62-1.83 | 53.80-2.00 | 47.47-2.00 | 51.49-2.45 |
| No. reflections | 97,044 | 107,574 | 79,052 | 81,565 | 42,513 |
| $R_{work}$ | 16.19 (33.0) | 15.37 (23.60) | 18.34 (22.70) | 15.26 (21.30) | 17.42 (26.0) |
| $R_{free}$ | 18.87 (33.80) | 17.83 (26.00) | 21.81 (27.40) | 18.38 (24.60) | 21.23 (30.20) |
| No. atoms | | | | | |
| Protein | 6652 | 6669 | 6657 | 6653 | 6650 |
| Ligand/ion | 272 | 329 | 256 | 303 | 339 |
| Water | 668 | 776 | 773 | 633 | 290 |
| B-factors (Å$^2$) | | | | | |
| Protein | 31.46 | 24.01 | 17.81 | 20.74 | 28.87 |
| Ligand/ion | 59.56 | 56.20 | 49.34 | 43.67 | 54.76 |
| Water | 38.83 | 34.38 | 27.34 | 30.58 | 28.29 |
| R.m.s. deviations | | | | | |
| Bond lengths (Å) | 0.009 | 0.008 | 0.009 | 0.007 | 0.007 |
| Bond angles (°) | 1.407 | 1.327 | 1.366 | 1.297 | 1.346 |
| Ramachandran | | | | | |
| Favored (%) | 97.96 | 97.74 | 97.72 | 97.24 | 97.74 |
| Allowed (%) | 2.04 | 2.14 | 2.28 | 2.64 | 2.14 |
| Disallowed (%) | 0 | 0.12 | 0 | 0.12 | 0.12 |

[a]One single crystal was used for each data collection
[b]Values in parentheses are for the highest-resolution shell

conformation in the substrate-binding subsite −1[30] and is stabilized by hydrogen bonds to the side-chains of D404, D518, R600, D616, and H674. Further stabilizing interactions are provided by hydrophobic contacts with W376, I441, W516, M519, W613, and F649 (Fig. 3a, b). Apart from a 20° tilt in the side-chain of W376, no major conformational changes occur upon DNJ binding in the rhGAA active site. This is not the case for binding of NHE-DNJ, which adopts the same pose as DNJ, but where the hydroxyethyl substituent induces substantial conformational changes to the side chains of M519 and W481 (Fig. 3c, d). In this ligand-induced conformation, rhGAA residue W481, located on the tip of insert I, establishes a stabilizing interaction with the hydroxyethyl substituent of NHE-DNJ. Similar conformational changes can be observed upon binding of NHE-DNJ to the N-terminal subunit of MGAM (NtMGAM)[31] (Supplementary Fig. 3a).

**Substrate recognition and specificity.** To gather a perception of substrate recognition by rhGAA, we acquired the diffraction data extending to 2.45 Å resolution for a rhGAA crystal soaked with acarbose, a non-cleavable α−1,4-tetrasaccharide substrate analogue (Table 1). In this complex, the valienamine ring lodged in subsite −1 adopts a $^2H_3$ half-chair conformation, but despite this conformational difference with respect to DNJ bound in the $^4C_1$ half-chair conformation, the hydrogen-bonding pattern in subsite −1 is essentially identical for the two compounds (Fig. 3e, f and Supplementary Fig. 3b). The non-hydrolysable "interglycosidic" nitrogen occupies the catalytic centre and is hydrogen

bonded to the catalytic acid/base D616. The 6-deoxyglucosyl moiety in subsite + 1 establishes via its 2- and 3-hydroxyl groups hydrogen bonds with the conserved R600 and with D282 originating from a loop in the N-terminal β-sheet domain. Though not conserved in sequence, residues of this loop interact invariably with carbohydrate ligands in all the available crystal structures of family GH31 members. The maltose unit of acarbose in subsites + 2 and + 3 does not make any direct interactions with rhGAA and is rather stabilized by crystal lattice packing interactions and a water mediated contact with the side chain of W618, located at the rim of the substrate-binding pocket. This paucity of interactions suggests that rhGAA possesses only two productive substrate-binding sites, subsites −1 and + 1, but interestingly, the acarbose maltose moiety bound to rhGAA adopts the same overall pose as when bound to NtMGAM[22] suggesting a functional role also for subsites + 2 and + 3 in rhGAA (Supplementary Fig. 3c).

Glycogen is a large polymer build-up by linear chains of α−1,4-linked glucose residues carrying α-1,6-linked branches. In the cytosol the energy storage particle is degraded by the concomitant action of glycogen phosphorylase and a debranching enzyme. Within the lysosome, no debranching enzyme is present and GAA must assure the hydrolysis of both α-1,4- and α-1,6-glycosidic linkages. However, rhGAA shows clear preference for the former linkage, as the specificity constant on maltose is 32-fold higher when compared to the specificity constant on isomaltose (Supplementary Table 2). To uncover the structural basis for the dual substrate specificity, we attempted soaking of

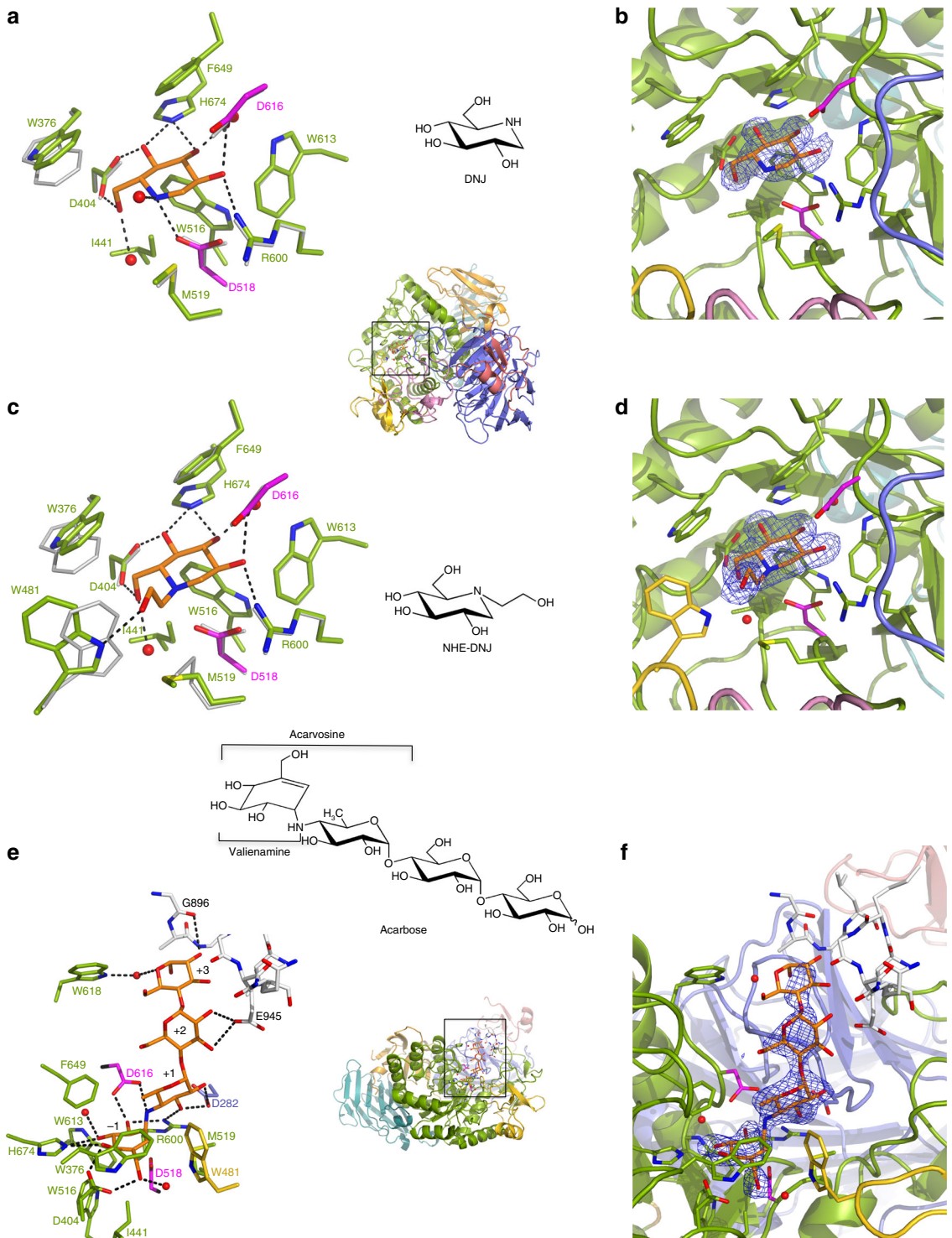

**Fig. 3** Ligand binding to rhGAA. DNJ (**a**), NHE-DNJ (**c**) and acarbose (**e**) colored in orange bound to rhGAA, color-coded as in Fig. 2c, overlapped onto unbound rhGAA (grey) in (**a**) and (**c**). Hydrogen bonding interactions are represented as dashed lines. In **e**, substrate-binding subsites are numbered and residues of a symmetry-related molecule are depicted in white sticks. Unbiased $F_o$–$F_c$ difference electron density maps, calculated before incorporation of the ligands into the models and contored at 3.0 σ, are shown in **b**, **d** and **f**

rhGAA crystals with the non-hydrolysable tetrasaccharide glucopyranosyl-α-(1,6)-thio-maltotriose. However, this approach failed, most likely due to crystal contacts impeding accommodation of the tetrasaccharide, which by virtue of the α-1,6-linkage is slightly longer than acarbose. The latter has been observed to barely fit into the substrate-binding cleft and closely abut a

symmetry related molecule (Fig. 3e). We derived an α-1,6-linked disaccharide bound within the active site of rhGAA by a structural superposition of rhGAA with the catalytic $(\beta/\alpha)_8$ domain of *Blautia obeum* GAA in complex with isomaltose[32] (rmsd of 1.50 Å for 318 aligned Cα positions). The model reveals that no steric hindrance occurs by the accommodation of

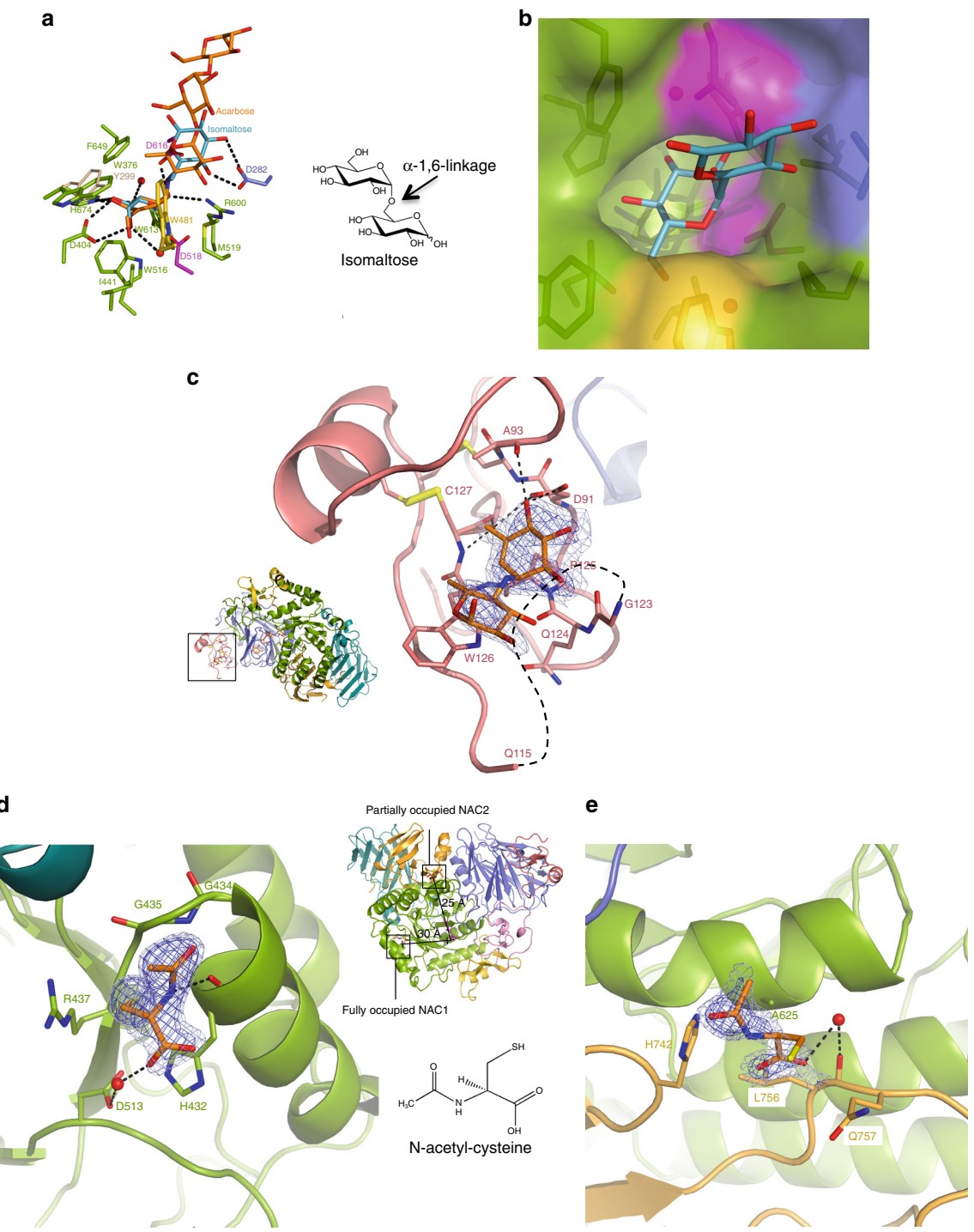

**Fig. 4** rhGAA substrate recognition and specificity and allosteric chaperone binding sites. **a** Model of isomaltose (steelblue), derived from an overlap with *B. obeum* α-glucosidase in complex with isomaltose (PDB ID 3MKK), superposed onto the rhGAA-acarbose complex (rmsd of 1.50 Å for 318 aligned Cα positions). **b** Model of isomaltose bound to rhGAA in surface representation. **c** The secondary substrate-binding site of rhGAA with unbiased $F_o$–$F_c$ difference electron density map calculated before incorporation of acarvosine (orange) into the model and contoured at 2.0 (lightblue) and 3.0 (blue) σ. The surface loop removed by proteolytic treatment is represented by dashed lines. **d** NAC1 (orange) in the fully occupied binding site. **e** NAC2 (orange) in the partially occupied binding site. In **d** and **e**, unbiased $F_o$–$F_c$ difference electron density maps, calculated before incorporation of NAC into the model, are shown in lightblue (2.0 σ) and blue (3.0 σ)

a α-1,6-linkage between subsite −1 and + 1 and that productive hydrogen bonds between the glycopyranose moiety in subsite + 1 and D282 are maintained (Fig. 4a, b). However, due to the increased length of the α-1,6-linkage with respect to an α-1,4-linkage, the stabilizing hydrogen bond with R600 is weakened,

which might explain lower activity of the enzyme on α-1,6-linked branches.

**The secondary substrate-binding domain**. In the rhGAA-acarbose complex, we could observe the acarvosine moiety of a

second acarbose molecule lodged in a pocket within the N-terminal trefoil Type-P domain, situated on the same face of rhGAA where the active site is located and 25 Å away from acarbose bound therein (Fig. 4c). The valienamine moiety hydrogen-bonds with its 4- and 6-hydroxyl groups to the main-chain nitrogen and oxygen of C127, the main-chain oxygen of A93 and the side-chain of D91. Further stabilizing interactions are provided by stacking interactions with P125 and W126. P125 and D91 are conserved or replaced by conservative substitutions in the family GH31 enzymes comprising a trefoil Type-P domain (Supplementary Fig. 4a). Adjacent to the secondary carbohydrate-binding site the surface loop G116-M122 had been trimmed by proteolytic treatment. Conceivably, the removal of the segment G116-M122, unique to the lysosomal representatives of family GH31 (Supplementary Fig. 4a, b), might contribute to the formation of the secondary substrate-binding pocket. One could envisage that the secondary acarbose-binding site on rhGAA represents a genuine substrate-binding site possibly enhancing the processivity of the enzyme. This site could also represent a pharmacophore for in silico screening for novel pharmacological chaperones. Most notably, a trisaccharide derived from acarbose is bound to the N-terminal domain of family GH31 α-1,4-glucan lyase from *Gracilariopsis lemaneiformis*[33] in a position close to the secondary substrate-binding site described here (Supplementary Fig. 5).

**N-acetylcysteine is an allosteric pharmacological chaperone.** The known pharmaceutical drug N-acetylcysteine (NAC) has been shown to act as allosteric pharmacological chaperone enhancing the stability of mutated endogenous GAA and rhGAA used for ERT, without affecting the catalytic activity[18]. To harness the therapeutic potential of NAC for Pompe disease via structural studies, we pursued the crystal structure of rhGAA in complex with NAC. The 1.83 Å resolution structure of the complex obtained by crystal soaking experiments (Table 1), reveals two NAC-binding sites remote from the active site (Fig. 4d, e), providing structural evidence that NAC is indeed an allosteric chaperone, as anticipated by functional studies[18]. We ascertained via activity assays and differential scanning fluorimetry that NAC stabilizes equally well rhGAA and the proteolytically matured enzyme produced in this study, and that the chaperone is specific towards GAA, as NAC has no stabilizing effect on a GH31 homologue from rice (Supplementary Fig. 6a, b and Supplementary Tables 3 and 4). In the rhGAA-NAC complex structure a NAC molecule, designated as NAC1, is located about 30 Å away from the active site, at the interface between the $(\beta/\alpha)_8$ barrel and the distal β-sheet domain (Fig. 4d). NAC1 binds to the main chain carboxyl of H432 via its amide nitrogen and the carboxyl function makes a water mediated contact with the side chain of D513. The Cβ-Sγ and the acetyl groups establish stacking interactions with the guanidine group of R437 and the loop G434-G435, respectively. A second, partially (75%) occupied NAC molecule (NAC2), is lodged at the interface between the $(\beta/\alpha)_8$ barrel and the proximal β-sheet domain at a distance of 25 Å from the active site (Fig. 4e). The carboxyl function of NAC2 makes a water-mediated contact with the main-chain oxygen of L756, the thiol stacks against the side chain of Q757 and the acetyl group stacks against the side chain of H742. The two NAC molecules establish fewer and weaker interactions with rhGAA than the chaperones DNJ and NHE-DNJ. This reflects the higher concentrations of NAC needed for chaperone activity (mM vs, μM) as exemplified by the $K_D$ of 11.57±0.74 mM that we obtained by differential scanning fluorimetry and by the $K_i$ of 3.4 and 3.0 μM for DNJ and NHE-DNJ, respectively[18,29]. A sigmoidal saturation curve best fitting the experimental points supports the

NAC allosteric fully (NAC1) and partially (NAC2) occupied binding sites (Supplementary Fig. 6c). The stacking interactions established between the acetyl groups of NAC1 and NAC2 and rhGAA must contribute significantly to the binding energy, since amino acids related to NAC, such as N-acetylserine and N-acetylglycine have equally a stabilizing effect on rhGAA, whilst the non-acetylated counterparts have no effect[18]. The structure of the rhGAA-NAC complex shows a decrease in atomic thermal displacement parameters of residues surrounding the NAC-binding sites with respect to that of unbound rhGAA, pointing toward an overall inter-domain stabilizing function driven by NAC (Supplementary Fig. 7a–d). In the crystal NAC appears also to exercise an anti-oxidative effect as residue C938 is oxidized to the sulfenic acid form in all the crystal structures described here, except for the structure of the rhGAA-NAC complex (Supplementary Fig. 8a, b).

**NAC and iminosugars show different chaperoning profiles.** Some mutations leading to Pompe disease are responsive to both NAC and the iminosugars DNJ and NB-DNJ, while others are targeted specifically by one or the other pharmacological chaperone[16–18]. The vast majority of GAA mutants responsive to the chaperone activity of iminosugars are located either in reach of the active site or on structural elements contributing to its global architecture. In Pompe patient, fibroblasts and mouse models the GAA mutants most responsive to NAC (A445P, Y455F, and L552P)[18] are located on inserts I and II, where the corresponding wild-type residues contribute to domain stabilizations. A445 together with F487 defines the boundaries of insert I, and the A445P mutant most likely destabilizes the entire scaffold of insert I (Supplementary Fig. 9a). The side-chain hydroxyl of Y455 on insert I interacts with a long loop of the catalytic domain whilst the side chain of L552 on insert II makes hydrophobic interactions with residues from the N-terminal β-sheet domain (Supplementary Fig. 9b, c). The respective Y455F and L552P mutants have most certainly an overall destabilizing effect, which is clearly rescued by NAC. Hence, the chaperone action of iminosugars can be considered as a conformational rescue of structural perturbed active sites, whereas the effect of the allosteric chaperone NAC appears to be due to the stabilization of overall or local conformational fluctuations.

## Discussion

Although lysosomal storage diseases have low incidence in overall population, they represent a heavy burden to patients' health and have an important socio-economic impact. ERT with recombinant human rhGAA has become available as drug for the standard care for Pompe disease a decade ago and has proven to be beneficial for patient's survival and to stabilize the disease course[5–8]. Undoubtedly, structural information was desired to harness the therapeutic value of rhGAA, but remarkably, this live-saving drug could not be crystallized so far and its 3D-structure remained elusive. Yet, shortly after submission of this manuscript the Garman group released two isomorphous structures of recombinantly expressed human GAA in complex with glucose at the Protein Data Bank, with accession numbers 5KZW and 5KZX.

In this study we employed in situ proteolysis to afford well-diffracting crystals of rhGAA and interestingly the crystallographic model we obtained corresponds to the mature forms of recombinant and placental GAA found in the endosome/lysosomal compartment[20]. One of the peptide segments removed by proteolytic treatment (G116-M122) is specific to the lysosomal representatives of the GH31 enzyme family and its removal might be required for the generation of the secondary substrate-binding

site in the trefoil Type-P domain. De facto, none of the over a dozen native or complex structures of the human intestinal homologues MGAM and SI, where the peptide range corresponding to rhGAA G116-M122 is present, contains a carbohydrate or carbohydrate-like ligand such as glycerol in the area corresponding to the rhGAA secondary substrate-binding site. The secondary substrate-binding site might favor the adherence of the enzyme onto the glycogen particle during hydrolytic processing and enhance its ability to detangle the α-1,6-linked branch points, which in cytosolic glycogen metabolism are taken care of by a debranching enzyme, not present in the lysosome. Except for the human intestinal homologues, the other structurally characterized GH31 members do not contain a trefoil Type-P domain, but interestingly a secondary substrate-binding site is equally present in the N-terminal region of G. lemaneiformis α-1,4-glucan lyase, which degrades starch following an elimination mechanism instead of hydrolysis. With an increasing amount of forthcoming structural and functional data it should be interesting to see whether within glycoside hydrolase family GH31 relationships can be established between particular modes of substrate binding and degradation.

Despite the clinical benefits of ERT with rhGAA, patient's response is very variable and mostly limited by insufficient enzyme targeting to the lysosomal compartment, high cost and heavy burden of treatment. In recent years PCT, presenting advantages such as low cost and burden, and high distribution volume, has emerged as an attractive alternative, either alone or in combination with ERT, for the treatment of LSDs[14,34]. For Pompe disease, for which clinical trials combining ERT and treatment with NB-DNJ and DNJ have recently been concluded[28,35], PCT is particularly promising as pharmacological chaperones are small molecules that are expected to penetrate membranes and reach therapeutic levels in target tissues, including skeletal muscle (38–55% of human body weight), more easily than rhGAA. A potential drawback of most pharmacological chaperones under study for the rescue of lysosomal glycosidases is that they are active site-directed iminosugars and inhibitors of the target enzymes. Furthermore, glucose configured iminosugars targeting GAA have the additional drawback of inhibiting also the intestinal starch-digesting enzymes and the endoplasmic reticulum glycoprotein processing enzyme GAA II. The identification of second-generation chaperones able to stabilize target enzymes without inhibitory action and undesired off-target effects is thus highly desirable. In this study, we were able to harness by structural and biochemical analyses the therapeutic value of the previously identified pharmacological chaperone NAC. On the basis of docking studies using a structural model of GAA derived from the X-ray structure of NtMGAM, the authors of the original report on the allosteric chaperoning function of NAC[18] proposed a NAC-binding site at the interface between the catalytic (β/α)$_8$ barrel and the distal β-sheet domain of GAA. Although located in proximity of the observed NAC1 binding site, this "virtual" pocket derived from a homology-based model is not present in the experimental 3D structure of rhGAA, emphasizing that homology models can provide guidance to virtual screening up to a certain extent, but ultimately experimental data are needed for precise mapping of binding sites. Interestingly, an astonishingly high number of small solvent molecules originating from the crystallization medium (glycerol, ethylene glycol, sulphate, chloride) are present in the different rhGAA structures presented here. Based on the principle of the multiple solvent crystal structures approach[36], the different ligand binding sites might provide guidance to pharmacophore mapping in high-throughput screening experiments devoted to the identification of novel allosteric chaperones. We envisage that the structural framework provided here will allow mapping of disease

mutations and guide clinicians in therapeutic choices and furthermore supply an accurate molecular scaffold for in silico screening to map novel druggable sites and identify novel pharmacological chaperones with complementary and synergistic effect on adverse GAA mutations.

## Methods

**Sample preparation.** rhGAA (alglucosidase alfa, Myozyme®) produced in CHO cells in the precursor form was from Genzyme, Cambridge, MA. As source of enzyme, the residual amounts of the infusions of Myozyme® administrated for the treatment of Pompe patients at the Department of Translational Medical Sciences, Federico II University, Naples, Italy, were used. For the infusions, Myozyme® was conditioned in a buffer composed of 110 mM mannitol, 36 μM polysorbate 80 (Tween80), 3 mM $Na_2HPO_4 \bullet 7H_2O$ and 21 mM $NaH_2PO_4 \bullet H_2O$, pH 6.1. One aliquot of the infusion residuals was dialyzed against a buffer composed of 10 mM HEPES pH 7.0 and 150 mM NaCl, whereas a second aliquot was dialyzed against 10 mM Na-citrate buffer pH 5.5, supplemented with 150 mM NaCl. Extensive crystallization trials with rhGAA in the original buffer and in the HEPES and citrate buffer conditions provided crystals diffracting at the best to about 7 Å resolution. Protein disorder prediction with the computational tool DisEMBL[21] (http://dis.embl.de/) indicated the existence of disordered regions within the rhGAA sequence and limited proteolysis was thus envisaged to remove potential disordered surface loops deleterious for the formation of well-ordered crystals. To this aim, rhGAA was dialyzed against a buffer composed of 10 mM HEPES pH 7.6, 200 mM NaCl and 2 mM $CaCl_2$ and α-chymotrypsin and subtilysin were tested at different protease to protein mass ratios ranging from 1:2 to 1:100, at different incubation times and incubation temperatures. At selected time points, aliquots were removed from the reaction mixture and analysed by SDS-PAGE. Best and reproducible results were obtained by α-chymotrypsin treatment overnight at 20 °C and with a protease/rhGAA mass ratio of 1:2, affording a band on SDS-PAGE (8%) with a mass difference of ~5 kDa with respect to the rhGAA precursor (Fig. 2a). The proteolysed enzyme was separated from the proteolytic reaction mixture by size-exclusion chromatography in 10 mM HEPES pH 7.0 and 100 mM NaCl using a HiLoad 16/60 Superdex 200 column, and concentrated to approximately 3.7 mg ml$^{-1}$ using a centrifugal filter unit with a 30-kDa cutoff. N-terminal sequencing of proteolysed rhGAA was performed using a PROCISE® 494 sequencing system.

**Activity tests on the rhGAA forms.** The activity assays were performed at 37 °C by adding 5 μg of precursor or mature forms of rhGAA to 0.2 ml of reaction mixture containing 20 mM 4NP-Glc and 100 mM Na-acetate pH 4.0. After 2 min the reactions were blocked by adding 0.8 ml of 1 M Na-carbonate pH 10.2. Absorbance was measured at 420 nm at room temperature and the extinction coefficient to calculate enzymatic units was 17.2 mM$^{-1}$ cm$^{-1}$. One enzymatic unit is defined as the amount of enzyme catalyzing the conversion of 1 μmol substrate into product in 1 min, under the indicated conditions. Spontaneous hydrolysis of the substrates was subtracted by using appropriate blank mixtures without enzyme. All the kinetic data were calculated as the average of at least two experiments and were plotted and refined with the program Prism 5.0 (GraphPad Software, San Diego, CA USA).

**Crystallization and structure solution.** Crystallization trials were performed by the sitting-drop vapour-diffusion method using a Mosquito® (TTP labtech) crystallization robot and commercial screens. First crystal hits emerged from a condition of the Structure Screen 1 (Molecular Dimensions) containing 2.0 M ammonium sulphate, 0.1 M HEPES pH 7.5 and 2% v/v PEG400, and after several optimization rounds well diffracting crystals were obtained by the hanging-drop vapour-diffusion method by mixing 1 μl protein solution at 3.7 mg ml$^{-1}$ with 0.5 μl reservoir solution composed of 1.9 M ammonium sulfate, 0.1 M HEPES pH 7.0 and 2% v/v PEG400. Crystals belong to space group P2$_1$2$_1$2$_1$ with average unit cell axes of 97 × 103 × 128 Å and one molecule per asymmetric unit. Ligand soaking was performed by the addition of small amounts of powder of NAC, DNJ, NHE-DNJ or acarbose to the crystallization droplets (1–2 snowflake-like crystals of approximate dimensions of 0.5 mm), followed by incubation for ~1 h. Native rhGAA crystals were flash-cooled directly in liquid nitrogen, whereas crystals soaked with the various ligands were cryo-protected with reservoir solution supplemented with 15% (v/v) glycerol prior cooling. X-ray diffraction data were acquired at beam line Proxima1 at the Synchrotron Soleil, Gif-sur-Yvette, and at beam line ID23-2 at the European Synchrotron Radiation Facility, Grenoble. All diffraction data were processed with XDS[37] and scaled and merged using the CCP4[38] suite of programs Pointless, Aimless[39] and Truncate.

The structure of rhGAA was solved by molecular replacement with the program Phaser[40] using the coordinates of Homo sapiens NtMGAM (PDB ID 2QLY, 44% sequence identity with GAA) as search model (rotation function Z-score: 7.0; translation function Z-score: 21.4; number of clashes from packing analysis: 0; log likelihood gain: 1383). Subsequent auto-building with the program Buccaneer[41] yielded a model that was 93% complete with $R_{free}$ of 31.2%. Maximum-likelihood refinement, including TLS, and model adjustment were carried out with the programs Refmac[42] and Coot[43], respectively. Random sets of ~5% of reflections,

depending on the resolution limit, were set aside for cross-validation purposes. The composition of cross-validation data sets was systematically taken over from the parent data set. Model quality was assessed with internal modules of Coot[43] and with the Molprobity server[44]. Ligands were fitted into unbiased $F_o$–$F_c$ difference electron density maps calculated after 10 cycles of rigid body refinement with Refmac[42]. The data collection and refinement statistics are listed in Table 1 with representative electron density in Fig. 3, 4 and Supplementary Figs. 2 and 8. Figures representing structural renderings were generated with the PyMOL Molecular Graphics System (DeLano, W.L. The PyMOL Molecular Graphics on http://www.pymol.org/). Chemical structures were drawn with ChemDraw 9.0 Pro (Cambrigesoft Corp, Cambridge, MA). Sequence alignments were made using MUSCLE[45] and graphical rendering of the alignments, taking into account structural information, were made with ESPript[46].

**Differential scanning fluorimetry**. Thermal stability scans of rhGAA were performed as described in Porto et al., 2012[18]. Briefly, the enzyme was diluted 48-fold (0.1 mg ml$^{-1}$) in 25 mM Na-phosphate buffer, pH 7.4, 150 mM NaCl and 1:1000 SYPRO Orange dye (Life Technologies). Thermal scans were performed in triplicate in absence or in presence of 10 mM NAC, with steps of 1 °C min$^{-1}$ in the range 25–95 °C in a Real Time LightCycler (Bio-Rad). Fluorescence was normalized to the maximum value within each scan to obtain relative fluorescence. Melting temperatures were calculated according to Niesen et al., 2007[47]. The standard deviations for each melting temperature were calculated from three replicates.

The dissociation constant ($K_D$) of NAC was measured by thermal stability scans of rhGAA according to Vivoli et al., 2014[48]. DSF scans were performed as described above, in the range 0–28 mM NAC. The melting temperature values were plotted as function of ligand concentration. The experimental data were best fitted according to a simple cooperative model equation reported in Vivoli et al., 2014[48] by using the software GraphPAD Prism (GraphPad Software, San Diego, CA, USA). To measure the thermal stability of GAA from rice type V (resuspended in 2.8 M ammonium sulphate) (Sigma-Aldrich), the enzyme was diluted 34.5-fold (0.1 mg ml$^{-1}$) and the thermal scans were performed as described above for rhGAA.

**Kinetic constants of rhGAA on different substrates**. All substrates were purchased from Sigma-Aldrich. Kinetic constants on 4NP-Glc (1–35 mM) were measured as described above. Kinetic constants on maltose (1–150 mM) and isomaltose (15–200 mM) were measured in 100 mM Na-acetate buffer pH 4.0 at 37 °C, by using 5 and 15 µg of enzyme, in a final volume of 0.2 and 0.1 ml, respectively. After 2 min, the amount of glucose produced in the reactions was determined using a glucose oxidase-peroxidase system (D-Glucose kit, Megazyme). One unit of activity was defined as the amount of enzyme releasing 1 µmol of glucose per minute at the conditions described. Spontaneous hydrolysis of the substrates was subtracted by using appropriate blank mixtures without enzyme. All the kinetic data were calculated as the average of at least two experiments and were plotted and refined with the program Prism 5.0 (GraphPad Software, San Diego, CA, USA).

Kinetic constants on bovine and rabbit glycogens were measured by using the substrates ranging from 1 to 123 mg ml$^{-1}$ in 50 mM citrate-phosphate buffer pH 4.0 at 37 °C by using 5 µg of enzyme in the final volume of 0.1 ml. After 2 min, 0.1 ml of Na–carbonate 0.25 M was added to the reaction mixtures. To determine the amount of reducing sugars released 0.1 ml were analysed by the Somogyi–Nelson assay[49]. One unit of activity was defined as the amount of enzyme releasing 1 µmol of reducing equivalents per minute at the conditions described.

**Data availability**. The atomic coordinates and structure factors of rhGAA and complexes with NAC, DNJ, NHE-DNJ and acarbose have been deposited in the Protein Data Bank with accession numbers 5NN3, 5NN4, 5NN5, 5NN6 and 5NN8, respectively. All the other data are available from the corresponding author on request.

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

## Acknowledgements

We are grateful to S. Cottaz for providing glucopyranosyl-α-(1,6)-thio-maltotriose. We thank the European Synchrotron Radiation Facility (ESRF) and Synchrotron Soleil for beam time allocation and the beam line staff for assistance with data collection. This work was supported in part by the CNRS and the French Infrastructure for Integrated Structural Biology (FRISBI) ANR-10-INSB-05–01. Telethon support to G.P. is gratefully acknowledged.

## Author contributions

V.R.-Z. and S.G.: Performed limited proteolysis, purification and crystallization. G.S. performed crystallographic studies. R.I. and M.C.F.: Performed enzymology and Differential scanning fluorimetry studies, B.C.-P., M.M. and G.S: Designed the study. G.S.: Wrote the manuscript with valuable contributions from B.C.-P., Y.B., G.P. and M.M.

## Additional information

**Competing interests:** The authors declare no competing financial interests.

