## [Peer Review file · Nature Communications]

REVIEWERS' COMMENTS:

Reviewer #1 (Remarks to the Author):

This manuscript describes the long-awaited structure of human lysosomal acid-alpha-glucosidase (GAA), the enzyme mutated in Pompe disease. Despite availability of large quantities of pure enzyme produced for enzyme replacement therapy, a number of previous attempts to obtain good crystals had failed. The structure will be extremely valuable in helping to inform the development of new therapies for this rare but devastating lysosomal storage disease.

In addition to providing a high-resolution structure of excellent quality for this enzyme, the authors have contributed ligand-binding studies that will be of great value to those who wish to develop new therapies. Complexes with substrate analogues and with active-site directed inhibitors (deoxynojirimycin and its N-hydroxyethyl derivative) that stabilise the fold will contribute to the further development of typical pharmacological chaperones, i.e. those that improve the trafficking of active enzyme to the lysosome but at the cost of at least temporary inhibition of enzyme activity. A favourable alternative is to produce allosteric pharmacological chaperones that stabilise the enzyme without inhibiting it, and the structure of a complex with N-acetylcysteine demonstrates the mode of interaction with promising binding sites outside of the active site cleft.

The authors also make the interesting observation that a trefoil type-P domain appears to have a binding site that might anchor the enzyme on glycogen particles.

I was initially surprised and slightly disappointed that the authors have not provided an annotated list of all known disease-associated mutations or an image showing their locations on the structure. On reflection, I think that the combination of structure determination, several crystallographic binding studies, and investigations of thermal stability is probably enough for one paper. However, I look forward to seeing a detailed structural analysis of the effects of disease-associated mutations.

In a revised manuscript, the authors should mention that, shortly after submission of their manuscript, another group released two isomorphous structures of the same enzyme at the PDB, though there is no peer-reviewed publication accompanying those structures yet. The PDB entries 5kzw and 5kzx were released on 26 July 2017, but the only associated publication so far is an abstract from the WORLDSymposium 2016 meeting, published last year in *Molecular Genetics and Metabolism*.

There are a few minor corrections that would improve the manuscript.

In the introduction (e.g p.2 line 46 and p.3 line 56, the term "pharmaceutical chaperone" should be replaced by "pharmacological chaperone" for consistency with standard usage and with the rest of the manuscript.

In the discussion of N-acetylcysteine binding (p.8), it would help to have some idea of the concentration of N-acetylcysteine, so that the occupancies of the two sites can be understood in the context of the concentration dependence of the chaperone activity. Were the volume of the crystallization drop and the amount of powder added to it (p. 15) measured at least approximately?

On p.9, the term "thermal agitation factor" is appealing but it would be better to use something more standard, like "thermal displacement parameter" or "thermal motion factor".

On p.15 line 353, "scalded" should be "scaled".

The figures use attractive colours, but more saturated versions of the lighter colours should be

used for labels, some of which are almost invisible. In particular, it is very difficult to see the light blue label for the isomaltose in Figure 4a, and even the molecule is difficult to see in this colour.

In the supplementary material, the pairs of images in Supp Figs 2 and 7 don't appear to be stereos. Either the right and left images are identical, or the pinch angle is so small that there is no visible stereo effect.

Supplementary Figure 5 has not reproduced in the PDF version of the supplement, although what is in the small box is visible in the Word version of this document.

Randy J Read
Cambridge Institute for Medical Research
University of Cambridge

Reviewer #2 (Remarks to the Author):

The authors are congratulated on an interesting and clear story providing structural evidence for the activity of NAC as a stabilizer for hGAA. The following comments/suggestions should not delay publication significantly but would be helpful to consider.

1. Pages are not numbered but on Page 6 of the MS, how was the superposition onto the B. obeum enzyme carried out? What atoms were used and is there an rms deviation?
2. Acarvosine in the supplemental binding site: is this presumed to be acarbose where part is not visible? If not, where did it come from?
3. Figure 4 - bound compounds: The authors are commended for showing the 'unbiased' difference map. However, the density is not terribly convincing at 3-sigma. How about adding a lower contour level in another color?
4. B-factor stabilization: Fig S7: I don't follow the argument for 'correcting' the B-factors as described in the caption to Fig S7. Is this the same in effect as comparing the differences between the atoms in question and the average B-factor over all atoms for each structure? Also, if the compound is stabilizing the surrounding residues, should it not have a B-factor roughly matching those residues? Was the occupancy constrained to 1.0?
5. The goal is to stabilize the structure for activity in the lysosome. The stability measurements were done at pH7.4 (p. 16). Is the compound equally stabilizing at low pH? Similarly - may be quibbling - but the enzyme activity comparison between wild-type and mutant (Pg 4) was at what pH? pH 4.0 as in Methods?

Minor / typos:

Suggest the GH family (GH31) is stated explicitly in the intro, rather than implied later.

Pg 6, line 5 - "by virtue of"

Pg 15, line 10 should be "scaled"

SI Fig 4 caption, halfway down, should be "sucrase"

REVIEWERS' COMMENTS:

Reviewer #1 (Remarks to the Author):

This manuscript describes the long-awaited structure of human lysosomal acid-alpha-glucosidase (GAA), the enzyme mutated in Pompe disease. Despite availability of large quantities of pure enzyme produced for enzyme replacement therapy, a number of previous attempts to obtain good crystals had failed. The structure will be extremely valuable in helping to inform the development of new therapies for this rare but devastating lysosomal storage disease.

In addition to providing a high-resolution structure of excellent quality for this enzyme, the authors have contributed ligand-binding studies that will be of great value to those who wish to develop new therapies. Complexes with substrate analogues and with active-site directed inhibitors (deoxynojirimycin and its N-hydroxyethyl derivative) that stabilise the fold will contribute to the further development of typical pharmacological chaperones, i.e. those that improve the trafficking of active enzyme to the lysosome but at the cost of at least temporary inhibition of enzyme activity. A favourable alternative is to produce allosteric pharmacological chaperones that stabilise the enzyme without inhibiting it, and the structure of a complex with N-acetylcysteine demonstrates the mode of interaction with promising binding sites outside of the active site cleft.

The authors also make the interesting observation that a trefoil type-P domain appears to have a binding site that might anchor the enzyme on glycogen particles.

I was initially surprised and slightly disappointed that the authors have not provided an annotated list of all known disease-associated mutations or an image showing their locations on the structure. On reflection, I think that the combination of structure determination, several crystallographic binding studies, and investigations of thermal stability is probably enough for one paper. However, I look forward to seeing a detailed structural analysis of the effects of disease-associated mutations.

In a revised manuscript, the authors should mention that, shortly after submission of their manuscript, another group released two isomorphous structures of the same enzyme at the PDB, though there is no peer-reviewed publication accompanying those structures yet. The PDB entries 5kzw and 5kzx were released on 26 July 2017, but the only associated publication so far is an abstract from the WORLDSymposium 2016 meeting, published last year in *Molecular Genetics and Metabolism*.

There are a few minor corrections that would improve the manuscript.

In the introduction (e.g p.2 line 46 and p.3 line 56, the term "pharmaceutical chaperone" should be replaced by "pharmacological chaperone" for consistency with standard usage and with the rest of the manuscript.

In the discussion of N-acetylcysteine binding (p.8), it would help to have some idea of the concentration of N-acetylcysteine, so that the occupancies of the two sites can be understood in the context of the concentration dependence of the chaperone activity. Were the volume of

the crystallization drop and the amount of powder added to it (p. 15) measured at least approximately?

On p.9, the term "thermal agitation factor" is appealing but it would be better to use something more standard, like "thermal displacement parameter" or "thermal motion factor".

On p.15 line 353, "scalded" should be "scaled".

The figures use attractive colours, but more saturated versions of the lighter colours should be used for labels, some of which are almost invisible. In particular, it is very difficult to see the light blue label for the isomaltose in Figure 4a, and even the molecule is difficult to see in this colour.

In the supplementary material, the pairs of images in Supp Figs 2 and 7 don't appear to be stereos. Either the right and left images are identical, or the pinch angle is so small that there is no visible stereo effect.

Supplementary Figure 5 has not reproduced in the PDF version of the supplement, although what is in the small box is visible in the Word version of this document.

Randy J Read
Cambridge Institute for Medical Research
University of Cambridge

Reviewer #2 (Remarks to the Author):

The authors are congratulated on an interesting and clear story providing structural evidence for the activity of NAC as a stabilizer for hGAA. The following comments/suggestions should not delay publication significantly but would be helpful to consider.

1. Pages are not numbered but on Page 6 of the MS, how was the superposition onto the B. obeum enzyme carried out? What atoms were used and is there an rms deviation?
2. Acarvosine in the supplemental binding site: is this presumed to be acarbose where part is not visible? If not, where did it come from?
3. Figure 4 - bound compounds: The authors are commended for showing the 'unbiased' difference map. However, the density is not terribly convincing at 3-sigma. How about adding a lower contour level in another color?
4. B-factor stabilization: Fig S7: I don't follow the argument for 'correcting' the B-factors as described in the caption to Fig S7. Is this the same in effect as comparing the differences between the atoms in question and the average B-factor over all atoms for each structure? Also, if the compound is stabilizing the surrounding residues, should it not have a B-factor roughly matching those residues? Was the occupancy constrained to 1.0?
5. The goal is to stabilize the structure for activity in the lysosome. The stability measurements were done at pH7.4 (p. 16). Is the compound equally stabilizing at low pH? Similarly - may be quibbling - but the enzyme activity comparison between wild-type and mutant (Pg 4) was at what pH? pH 4.0 as in Methods?

Minor / typos:

Suggest the GH family (GH31) is stated explicitly in the intro, rather than implied later.

Pg 6, line 5 - "by virtue of"

Pg 15, line 10 should be "scaled"

SI Fig 4 caption, halfway down, should be "sucrase"

ANSWERS TO REVIEWERS' COMMENTS:

Reviewer #1 (Remarks to the Author):

This manuscript describes the long-awaited structure of human lysosomal acid-alpha-glucosidase (GAA), the enzyme mutated in Pompe disease. Despite availability of large quantities of pure enzyme produced for enzyme replacement therapy, a number of previous attempts to obtain good crystals had failed. The structure will be extremely valuable in helping to inform the development of new therapies for this rare but devastating lysosomal storage disease.

In addition to providing a high-resolution structure of excellent quality for this enzyme, the authors have contributed ligand-binding studies that will be of great value to those who wish to develop new therapies. Complexes with substrate analogues and with active-site directed inhibitors (deoxynojirimycin and its N-hydroxyethyl derivative) that stabilise the fold will contribute to the further development of typical pharmacological chaperones, i.e. those that improve the trafficking of active enzyme to the lysosome but at the cost of at least temporary inhibition of enzyme activity. A favourable alternative is to produce allosteric pharmacological chaperones that stabilise the enzyme without inhibiting it, and the structure of a complex with N-acetylcysteine demonstrates the mode of interaction with promising binding sites outside of the active site cleft.

The authors also make the interesting observation that a trefoil type-P domain appears to have a binding site that might anchor the enzyme on glycogen particles.

Response: We thank the reviewer for the positive and constructive feedback. In the revised manuscript we have taken into account the reviewer's comments and suggestions and our responses to the points raised are listed below:

I was initially surprised and slightly disappointed that the authors have not provided an annotated list of all known disease-associated mutations or an image showing their locations on the structure. On reflection, I think that the combination of structure determination, several crystallographic binding studies, and investigations of thermal stability is probably enough for one paper. However, I look forward to seeing a detailed structural analysis of the effects of disease-associated mutations.

Response: We thank the reviewer for this suggestion and have added to Supplementary a Table describing the structural consequences of missense mutations associated with Pompe disease.

In a revised manuscript, the authors should mention that, shortly after submission of their manuscript, another group released two isomorphous structures of the same enzyme at the PDB, though there is no peer-reviewed publication accompanying those structures yet. The PDB entries 5kzw and 5kzx were released on 26 July 2017, but the only associated publication so far is an abstract from the WORLDSymposium 2016 meeting, published last year in Molecular Genetics and Metabolism.

Response: We have added a sentence to the Discussion section mentioning that the Garman group has released two isomorphous structures of rhGAA at the Protein Data Bank shortly after submission of our manuscript. An overlap of these structures onto the unbound form of rhGAA described in our manuscript reveals that the structures are almost identical, with an rmsd of 0.19 Å for 844 aligned C α positions. Given this significant similarity we feel that a structural comparison would be obsolete in the context of this manuscript.

There are a few minor corrections that would improve the manuscript.

In the introduction (e.g p.2 line 46 and p.3 line 56, the term "pharmaceutical chaperone" should be replaced by "pharmacological chaperone" for consistency with standard usage and with the rest of the manuscript.

Response: We thank the referee for pointing out this language abuse and have corrected the mistake.

In the discussion of N-acetylcysteine binding (p.8), it would help to have some idea of the concentration of N-acetylcysteine, so that the occupancies of the two sites can be understood in the context of the concentration dependence of the chaperone activity. Were the volume of the crystallization drop and the amount of powder added to it (p. 15) measured at least approximately?

Response: We thank the referee for the suggestion and have added to the methods section that the amount of powder added to crystallizations droplets of ~1.5 μ l corresponded to "1-2 snowflake-like crystals of approximate dimensions of 0.5 mm". We refrain from going any further in quantification of "real" concentration of NAC within crystal solvent channels, as we feel that this would be too much of a speculation on density of snowflake-like crystals, diffusion rate, solvent accessibility within crystal and competition with small molecules present in the crystallization medium.

On p.9, the term "thermal agitation factor" is appealing but it would be better to use something more standard, like "thermal displacement parameter" or "thermal motion factor".

Response: We appreciate the reviewer's humour and kind suggestion and have replaced the term in question by a more standard annotation.

On p.15 line 353, "scalded" should be "scaled".

Response: We thank the reviewer for the thorough evaluation of the manuscript and have corrected the misspelling.

The figures use attractive colours, but more saturated versions of the lighter colours should be used for labels, some of which are almost invisible. In particular, it is very difficult to see the light blue label for the isomaltose in Figure 4a, and even the molecule is difficult to see in this colour.

Response: We thank the reviewer for the appreciation of the colouring scheme used to illustrate the diverse structural features of rhGAA and are grateful for the constructive suggestion. We have redrawn Figures 2, 3 and 4 and Supplementary Figures 3 and 9 (corresponding to numbering of revised version of Supplementary) using more saturated colours for labels and depiction of the isomaltose molecule in Figure 4a. We sincerely hope that in this new version the figures will be intelligible to the readers.

In the supplementary material, the pairs of images in Supp Figs 2 and 7 don't appear to be stereos. Either the right and left images are identical, or the pinch angle is so small that there is no visible stereo effect.

Response: We are grateful to the reviewer for pointing out our regrettable mistake made by reducing pinch angles when scaling images to A4 format. We have redrawn all stereo images (Supplementary Figures 2, 4 and 8) and verified the stereo effect on printed paper.

Supplementary Figure 5 has not reproduced in the PDF version of the supplement, although what is in the small box is visible in the Word version of this document.

Response: We thank the reviewer for pointing out this problem and apologize for not having carefully inspected the PDF version before submission of the manuscript. The problem was generated during the conversion process from Word to PDF. We have solved the problem.

Reviewer #2 (Remarks to the Author):

The authors are congratulated on an interesting and clear story providing structural evidence for the activity of NAC as a stabilizer for hGAA. The following comments/suggestions should not delay publication significantly but would be helpful to consider.

Response: We thank the reviewer for the encouraging and productive feedback. In the revised manuscript we have taken into account the reviewer's comments and suggestions and our responses to the points raised are listed below:

1. Pages are not numbered but on Page 6 of the MS, how was the superposition onto the *B. obeum* enzyme carried out? What atoms were used and is there an rms deviation?

Response: We thank the reviewer for pointing at the lack of this useful information and specified in the manuscript that the catalytic (β/α)₈ domain of *B. obeum* α -glucosidase in complex with isomaltose was superposed onto rhGAA, giving an rmsd of 1.50 Å for 318 aligned Ca positions).

2. Acarvosine in the supplemental binding site: is this presumed to be acarbose where part is not visible? If not, where did it come from?

Response: We thank the reviewer for this constructive comment and hope very much to have resolved the ambiguity by clearly stating that it was in the rhGAA-acarbose complex where we observed the acarvosine moiety of a second acarbose molecule.

3. Figure 4 - bound compounds: The authors are commended for showing the 'unbiased' difference map. However, the density is not terribly convincing at 3-sigma. How about adding a lower contour level in another color?

Response: We thank the reviewer for the suggestion and have redrawn Figure 4 by adding a lower contour level.

4. B-factor stabilization: Fig S7: I don't follow the argument for 'correcting' the B-factors as described in the caption to Fig S7. Is this the same in effect as comparing the differences between the atoms in question and the average B-factor over all atoms for each structure? Also, if the compound is stabilizing the surrounding residues, should it not have a B-factor roughly matching those residues? Was the occupancy constrained to 1.0?

Response: We could not simply compare the B-factors of individual residues of the rhGAA and rhGAA-NAC structures, as there subsisted an overall B-factor difference for protein atoms of 7.45 \AA^2 . This difference of overall B-factors of protein atoms is reflected by a difference of 7.26 \AA^2 of overall B-factors from Wilson plot. We considered that before B-factor comparison between unbound and NAC-bound structure, the addition of 7.45 \AA^2 to each individual B-factor of the rhGAA structure, in order to put B-factors approximately at the same scale, would be a more rigorous (although still only approximate) approach. The reviewer might have been misled by the colouring scheme: in Supplementary Figure 7a,c NAC molecules are not coloured according to B-factor, but according to atoms type, following the colouring scheme adopted in Figure 4. The overall B-factor of residues surrounding the primary NAC molecule as shown in Supplementary Figure 7a is 20.1 \AA^2 and the overall B-factor of residues surrounding the secondary NAC molecule as shown in Supplementary Figure 7c is 24.8 \AA^2 . The B-factors of the primary and secondary NAC molecules are 40.1 \AA^2 and 54.1 \AA^2 , respectively. These values correlate with overall B-factors of other ligand and solvent molecules. The occupancy of the primary NAC molecule was restrained to 1.0 and the occupancy of the secondary NAC molecule was restrained to 0.75.

5. The goal is to stabilize the structure for activity in the lysosome. The stability measurements were done at pH 7.4 (p. 16). Is the compound equally stabilizing at low pH? Similarly - may be quibbling - but the enzyme activity comparison between wild-type and mutant (Pg 4) was at what pH? pH 4.0 as in Methods?

Response: We thank the reviewer for the thoughtful considerations. The goal of PCT is to stabilize the structure for activity not only within the lysosome, but also during it's way to the lysosome, both rhGAA and the endogenous mutants. At pH 4.0, the physiological pH in the lysosome, the enzyme is active and stable and the effect of NAC as PC is not observed at low

pH. All the activity measurements were performed as indicate in Methods (in sodium acetate buffer pH 4.0).

Minor / typos:

Suggest the GH family (GH31) is stated explicitly in the intro, rather than implied later.

Response: We thank the reviewer for this wise suggestion and introduced the notion that GAA belongs to glycoside hydrolase family GH31 right at the beginning of the introduction.

Pg 6, line 5 - "by virtue of"

Response: We appreciate the help in approving the English of the manuscript and have corrected the fault.

Pg 15, line 10 should be "scaled"

Response: We thank the reviewer for the thorough evaluation of the manuscript and have corrected the misspelling.

SI Fig 4 caption, halfway down, should be "sucrase"

Response: We are grateful to the reviewer for spotting this mistake in the supplementary section, which pointed us towards the same mistake in the main manuscript. We have corrected both mistakes.